# Effect of Mg Addition on the Microstructure and Properties of a Heat-Affected Zone in Submerged Arc Welding of an Al-Killed Low Carbon Steel

**DOI:** 10.3390/ma14092445

**Published:** 2021-05-08

**Authors:** Yandong Li, Weiwei Xing, Xiaobing Li, Bo Chen, Yingche Ma, Kui Liu, Yi Min

**Affiliations:** 1Key Laboratory of Extraordinary Bond Engineering and Advanced Materials Technology (EBEAM), Yangtze Normal University, Chongqing 408100, China; andyydlee@gmail.com; 2Institute of Metal Research, Chinese Academy of Sciences, No.72 Wenhua Road, Shenyang 110016, China; wwxing@imr.ac.cn (W.X.); bchen@imr.ac.cn (B.C.); ycma@imr.ac.cn (Y.M.); kliu@imr.ac.cn (K.L.); 3School of Metallurgy, Northeastern University, Shenyang 110819, China; miny@mail.neu.edu.cn

**Keywords:** Al-deoxidized low carbon steel, Mg–Al–O inclusion, heat-affected zone (HAZ), acicular ferrite, submerged arc welding

## Abstract

To reveal the effect of Mg treatment on the microstructure evolution behavior in the actual steel welding process, the microstructure and properties of Al-deoxidized high-strength ship plate steel with Mg addition were analyzed after double-side submerged arc welding. It was found that the Al–Mg–O + MnS inclusion formed under 26 ppm Mg treatment could promote acicular ferrite (AF) nucleation in the coarse-grained heat-affected zone (CGHAZ) and inhibit the formation of widmanstätten ferrite and coarse grain boundary ferrite. In the fine-grained heat-affected zone (FGHAZ) and intercritical heat-affected zone (ICHAZ), polygonal ferrite and pearlite were dominant. Al–Mg–O+MnS cannot play a role in inducing AF, but the grain size of ferrite was refined by Mg addition. The impact toughness in HAZ of the Mg-added steel was higher than that of Mg-free steel. With the heat-input rising from 29.55 to 44.11 kJ/cm, it remained relatively stable in Mg-treated steel. From the fusion line to the base metal, the micro-hardness of the fusion zone, CGHAZ, ICHAZ and FGHAZ decreased to some extent after Mg addition, which means the cold cracking tendency in the welding weak zone could be reduced. Finally, the mechanisms of Mg-containing inclusion-induced AF were also systematically discussed.

## 1. Introduction

High-strength ship plate steel is widely used in military ships, special vessels, large ships, and other major national defense and civil-use fields. In the hull structure forming process, welding is an unavoidable working procedure. With the demand for both ship plate thickness increments and welding cost reductions, higher input energy is crucial to improve the welding efficiency, which in turn puts higher requirements on the weldability of the plate steels. Normally, the heat-affected zone (HAZ) from the weld joint to the base metal includes a fusion zone, coarse-grained heat-affected zone (CGHAZ) with a thickness of about 1.5 mm, fine-grained or recrystallization heat-affected zone (FGHAZ) with a thickness of about 1.0 mm, and an intercritical heat-affected zone (ICHAZ) with a thickness of about 2.0 mm [1]. The CGHAZ is very close to the weld seam with high peak temperature and obvious coarsening of the prior austenite grain size during welding, which often becomes the weakest zone of the ship plate steel and seriously affects the service reliability [2].

Oxide metallurgy is an effective method proposed in recent years to improve the performance of the steel welding heat-affected zone [3,4]. This method induces a large amount of small and dispersed oxide particles formed in steel through an appropriate deoxidization process. During the post-weld cooling process, these oxide particles can be used as ferrite nucleation sites to promote the formation of interlaced refine acicular ferrite (AF) which refines the heat-affected zone microstructure and improves its mechanical performance [3,4]. Among the various deoxidization processes, Mg treatment has always been a research focus in the oxide metallurgy field [5]; because Mg and oxygen have a strong thermodynamic affinity, the Mg-containing oxide particles are not easy to aggregate [6], and easily promote the formation of AF [7,8]. In recent years, numerous studies have been conducted both in China and globally focusing on the role of Mg treatment in the steel welding process. The typical related works are listed and summarized in Table 1 [4,9,10,11,12,13,14,15,16,17,18,19,20,21,22,23]. As can be seen from this table, currently, the influence of Mg treatment on steel welding performance is commonly analyzed by welding thermal simulation experiments and Mg-bearing inclusions are mainly MgO, MgS, and Ti–Mg oxides. Little attention has been paid to xMgO·Al_2_O_3_ (cubic structure) inclusion produced by Al deoxidization, which is quite common in industry processing. In addition, as can be seen from Table 1, with the continuous in-depth exploration on Mg treatment process, there has been some research concerning the effect of Mg treatment on the actual welding behavior of steel rather than the thermal simulation test. For instance, Li et al. [18] studied the effect of Ti–Mg–Al–O inclusions on the microstructure and properties of HAZ in actual steel plate welding with thicknesses of 16 mm, 25 mm and 40 mm. For xMgO·Al_2_O_3_, our previous research [20] systematically explored its effect on the microstructure and property evolution of a high-strength ship plate steel under thermal simulation test, but the actual evolution behavior still needs to be further studied under the real welding process.

Submerged arc welding is often used in actual manufacturing procedures to improve efficiency of the hull structure forming. In this research, this method was also adopted to analyze the effect of Mg treatment on F-Grade high-strength ship plate steel. The microstructure and property evolution of the fusion zone, CGHAZ, FGHAZ and ICHAZ were systematically investigated to explore the effect induced by trace Mg addition in the actual welding process. The possible mechanisms of AF nucleation by Mg-containing oxide were discussed based on the Mn-depleted zone (MDZ) theory and the minimum disregistry theory.

## 2. Materials and Methods

### 2.1. Materials Preparation

The experimental steels were smelted in a 30 kg vacuum induction furnace (Vacuum Research Institute, Shenyang, China). Considering that Mg has high vapor pressure at steelmaking temperatures, after adding Fe–Al alloys, Ni–Mg alloy instead of pure Mg was added into the molten steels with the inside pressure of the furnace maintained at −0.03 MPa by Ar gas blowing. The contents of O and N were determined with a TCH-600 oxygen–nitrogen analyzer (LECO Company, St. Joseph, MI, USA), and the spectral analysis results of the steel ingots are compiled in Table 2, where No.1 is the benchmark steel, and No.2 is the Mg-treated steel. After cut head and tail, the ingots were forged and rolled into 13 mm steel plates; the detailed procedure can be found elsewhere [20]. The rolled microstructures of these two steels are shown in Figure 1, which shows that the Mg-free steel mainly includes plenty of polygonal ferrites (PF) and a small amount of bainitic ferrite (BF) and pearlite (P). With Mg addition, a large amount of AF is formed instead of PF. The mechanical properties of these two steel plates are as follows: yield strength 448 MPa (No.1), 473 MPa (No.2); tensile strength 545 MPa (No.1), 605 MPa (No.2); elongation 33.6% (No.1), 36.5% (No.2), values which were also reported in our previous paper [7]. Additionally, average longitudinal impact toughness (−60 °C) was measured as 246 J (No.1), 261 J (No.2).

### 2.2. Submerged Arc Welding

The welding in this research was carried out on an MZ-1000 submerged arc welding testing machine (Panasonic, Japan) (Figure 2). The welding groove was I-type, the welding wire was Ø4 mm H10Mn2, and the weld flux was SJ101; the chemical compositions are presented in Table 3 and Table 4, respectively. Double-side welding was adopted, i.e., after one side (named front-side) was welded, the other side (named back-side) was cleaned and then welded. The welding heat input was kept constant on the front side and changed into different levels on the back side. No preheating (before welding) nor heat treatment (after welding) was implemented. The welding line energy q/v in the welding procedure can be calculated by Equation (1). The experimental steel plates were relatively thinner than the engineered steel plates; therefore, when the heat input applied to the back-side was more than 44 kJ/cm, the steel plate was burned through. The welding parameters set in this research are shown in Table 5. The heat input to the front side was kept at the same level (about 20 kJ/cm). Therefore, in this paper, the effect of welding heat input on the microstructure and properties of HAZ is based on the difference of the back-side line energy.
(1)$$q/v=\eta \mathrm{UI}/v\left(\mathrm{kJ}/\mathrm{cm}\right)$$where U (V) is the welding voltage, I (A) is the welding current, v (cm/s) is the welding speed, and η is the thermal efficiency coefficient; for submerged arc welding, the thermal efficiency coefficient is 1.0 [18].

### 2.3. Mechanical Properties Test and Microstructure Characterization

To specify the position of the HAZ in the plates with different welding line energies, the welding plate was first etched to locate the boundary between the base steel and HAZ, as the dotted line shows schematically in Figure 3. Standard Charpy V-notch impact specimens (10 mm × 10 mm × 55 mm) were prepared from the rectangle marked position according to GB/T229-2007. The V-notch was set at the I-site, located in the HAZ region, which was determined by the etched results. As can be seen from this figure, the HAZ in the after-welded steel plate is in a curved shape (dotted line); therefore, it might be difficult to guarantee that the V-notch completely passed through a certain area in the HAZ. Despite this, in this research, the V-notch was made close to the edge of the welding seam which was through the HAZ. The impact test was conducted by a ZBC2502-D Mattes pendulum impact tester, and the impact energy was 500 J·cm^−2^. The impact test was performed at −60 °C, and the result was the average value of three samples; the standard deviation (SD) of the test values was also calculated. The impact fracture morphology was observed with an SSX-550TM scanning electron microscope.

OLYMPUS-BX51 optical microscopy (OM) and IPP6.0 image analysis software (Media Cybernetics, Rockville, MD, USA) were used to analyze the size and number of non-metallic inclusions in the HAZ samples. The area of each selected view field was 92,750 µm^2^ and 64 continuous different view-fields were analyzed for each sample with a total area of 5.936 mm^2^. The inclusion morphology was observed and analyzed under Ultra Plus-Zeiss field emission scanning electron microscope (FESEM). FESEM and OM were also used to observe the microstructure of HAZ, which includes CGHAZ, FGHAZ and ICHAZ. Before observation, the samples were ground and polished first by standard method, and then slightly etched for 10 s with 3% nital solution to illustrate the detailed microstructure. The micro-hardness of the HAZ near to the fusion line in the samples processed by W2 procedure was measured by an HXD-1000TMC micro-hardness tester. The test load was 300 g, loading time was 10 s, and there were at least 10 test areas. The value for each area was the average of at least three measured points. The locations of the test area are indicated in Figure 4. A straight line was drawn perpendicular to the weld seam on the plate surface, and the test points were taken successively on this line. The microstructure of the CGHAZ in the heat-affected area changed significantly within a narrow range; the distance between test points in this area was only 0.25 mm, while in other areas the distance was 0.5 mm.

## 3. Results and Discussion

### 3.1. Non-Metallic Inclusions Characteristics in HAZ

Mg is a strong deoxidized element; in this steel, the total oxygen content was about 40 ppm (shown in Table 2), which made Mg mainly exist in the form of Mg-containing inclusions. To illustrate the effect of Mg on the microstructure and performance evolution behavior, the composition, morphology, size, and number of typical non-metallic inclusions in the heat-affected zone were firstly characterized. Figure 5 shows the SEM morphology and EDS analysis results of typical inclusions in the HAZ of Mg-free steel (Figure 5a) and Mg-treated steel (Figure 5b). It can be seen that the non-metallic inclusions in the HAZ of the two experimental steels are all complex inclusions with oxides in the central and MnS on the surface. The difference is, without Mg treatment, the endoplasmic oxide was mainly Al_2_O_3_, but with Mg treatment, the endoplasmic oxide was changed into Al-Mg-O with about 14 wt.% Mg.

The sizes and total number of the inclusions in HAZ are shown in Figure 6. As can be seen, with Mg addition, the size of inclusions was refined to a certain extent, i.e., the percentage of inclusions with size less than 2.5 μm increased from 90 to 98% (Figure 6a), and the corresponding total number rose from 1268 to 2204 (Figure 6b). In comparison with the inclusion characteristics of as-casted base metal reported in our previous research [8], it can be seen that the present thermal cycle in welding operation did not change the inclusion types for the two steels, while the percentage of inclusions with size below 2.5 μm appeared to be increased and the total number of inclusions also tended to be more after subjection to the welding process. The latest research results obtained by high-temperature laser confocal scanning microscope (CSLM) have confirmed that even a small amount of Mg addition could disaggregate the large-sized and clustered Al_2_O_3_ inclusions to fine and dispersed Al–Mg–O particles in only 15 s [6]. Hence, it is generally believed that the increase in the number and decrease in the size of the inclusions in Mg-treated steel is directly related to the modification of endoplasmic oxides from Al_2_O_3_ to Al–Mg–O.

### 3.2. Microstructure Characteristics in HAZ

Figure 7 shows the microstructure morphologies of the heat-affected zone, including the fusion zone, CGHAZ, FGHAZ and ICHAZ of the benchmark steel under W1 and W2 welding procedures, respectively. It can be identified that, for the steel without Mg treatment, the microstructures near the fusion line were mainly granular bainite ferrite (GF), widmanstätten ferrite (WF) and coarse grain boundary ferrite (GBF) for both W1 and W2 process conditions (Figure 7a,e). When the welding input energy is increased, the volume fractions of WF and GBF in this area also increase.

Similar to the fusion zone, the peak temperature of CGHAZ is also relatively high, austenite grains usually grow larger, and the cooling rate is usually slow [2], thereby easily generating coarse GBF in the subsequent cooling process (as shown in Figure 7b,f). Moreover, with the welding heat input increased from 29.55 to 44.11 kJ/cm, the number of GBF increases. In FGHAZ, although austenite phase transition happens during the welding process, the peak temperature is not as high as CGHAZ, and austenite grains coarsening may not appear significant as CGHAZ [1]. In the subsequent cooling process, fine equiaxial ferrite grains and a small amount of pearlite are formed, as shown in the microstructure (Figure 7c,g). In ICHAZ, which is close to the base metal, austenite phase transition occurs in some areas of this zone during the welding process. Very fine and slightly larger grains can all be found in this region, i.e., the grain uniformity is relatively poor (Figure 7d,h). It can be distinguished that in both FGHAZ and ICHAZ, the grain sizes tend to coarsen with the increasing heat input energy (Table 6).

Figure 8 shows the microstructure morphologies of the heat-affected zone, including the fusion zone, CGHAZ, FGHAZ and ICHAZ of the Mg-treated steel under W1 and W2 welding procedures. By comparison, it is noted that after Mg treatment, only a few WF and GBF can be found, and no coarse WF and GBF are detected near the fusion line or in the CGHAZ, where the peak welding temperatures are very high (Figure 8a,b,e,f). The microstructures were determined as mainly bainite, including GF, acicular ferrite (AF) and bainitic ferrite (BF) for both W1 and W2 welding procedures. For FGHAZ and ICHAZ, the structure is similar to that of the benchmark steel, which is with fine equiaxial ferrite and a small amount of pearlite (Figure 8c,d,g,h). It needs to be pointed out that the grain size in the two areas is obviously finer than that of the benchmark steel under W1 and W2, and it seems more obvious in the W2 process. The corresponding microstructure characteristics under W1 and W2 welding conditions are compiled in Table 6.

### 3.3. Mechanical Properties of HAZ

#### 3.3.1. Impact Absorption Energy and Fracture Morphology

Table 7 shows the impact absorption energy change of the HAZ in No.1 and No.2 steel under welding process W1 and W2. It can be identified that for the two steels, with the increase in welding heat input from 29.55 to 44.11 kJ/cm, the average impact energies of HAZ are all decreased, with reduction rates of 13% and 6%, respectively. Meanwhile, it can also be seen that under the same welding procedure, the average impact energy of the steel with Mg is higher than that of the benchmark steel. All these results indicate that Mg treatment might promote performance of the experimental steel plate under the submerged arc welding.

Figure 9 shows the SEM morphology of impact fracture of No.1 and No.2 steel in crack-initiated area under W1 and W2 welding processes, respectively. It can clearly be observed that under the W1 and W2 welding processes, the impact fracture of No.1 and No.2 steel is typical dimple-microvoid accumulation ductile fracture. It can also be found that the depth of impact fracture dimples is shallower in No.1 steel than that of No.2 steel under both W1 and W2 welding conditions, which further shows that the Mg-added steel could possess a better impact toughness.

#### 3.3.2. Micro-Hardness

The Vickers hardness of welded joints can reflect the changes in microstructure and mechanical properties when they are under a welding heat cycle, and it is believed they play an important role in the mechanical properties test of welding process evaluation [24]. Normally, the region with higher hardness usually experiences higher strength, but lower plasticity and toughness, and also a tendency to cold cracking. Therefore, the strength, plasticity, toughness, and cold crack sensitivity of HAZ can all be generally estimated by measuring the hardness distribution of welding HAZ. The hardness changes from weld seam to base metal of No.1 and No.2 steels after W2 process are shown in Figure 10. As can be seen, the figure is divided into five parts: weld line (I), fusion zone and CGHAZ (II), FGHAZ (III), ICHAZ (IV), and base metal (V). The fusion zone is immediately next to the weld line and covers only a few grains in width, called the fusion line. Here, the fusion line is marked as the zero point of the horizontal axis. It can clearly be seen from Figure 10 that the micro-hardness in each weld joint region of the two experimental steels (noted as No.1 and No.2 in Figure 10) experiences the same changing behavior when after the same welding process. From the weld line to the base metal, the maximum hardness value is reached at the place between the fusion zone and CGHAZ, and then this value decreases obviously in the FGHAZ, while in ICHAZ, the hardness fluctuates greatly, and remains stable in the base metal. After Mg-treatment, the micro-hardness of the fusion zone, CGHAZ and FGHAZ slightly decreases. Therefore, it can be speculated that Mg treatment may have positive effect on reducing the cold cracking tendency to a certain extent.

### 3.4. Effect of GBF and WF Microstructure on the Toughness of HAZ

As analyzed above, for the Mg-free steel, both for W1 and W2 process conditions, some GBF and WF microstructures are formed in the fusion zone and CGHAZ, while only a few are detected in Mg-added steel. Normally, during the welding process, austenite grains are coarsened in the region with high peak temperature, which makes the ferrite tend to nucleate at the austenite grain boundary in the austenite–ferrite transition process and finally form coarse GBF and WF microstructures. Hence, these coarse microstructures are generally considered to have adverse effects on the toughness of CGHAZ [25,26].

Figure 11 shows the typical morphology of the secondary cracks along GBF and WF in the heat-affected zone of the Mg-free steel after the W2 process. It can be seen that the secondary crack directly propagates through the GBF (Figure 11a) and WF (Figure 11b) without any impeding effects. Therefore, it is reasonable to suggest that once a large amount of GBF and WF has formed in HAZ, the toughness will be greatly reduced. Thus, after submerged arc welding, the GBF and WF formed in the coarse grain zone of Mg-free steel will deteriorate mechanical properties. In the actual welding process, the width of the coarse grain zone is only about 1 mm (shown in Figure 10); it is hard to measure the impact toughness accurately. The impact energy tested in this paper would be more properly considered as the comprehensive impact toughness of different areas in the heat-affected zone. Nevertheless, the suppressing coarse GBF and WF nucleation becomes an important factor to improve CGHAZ toughness for the Mg-added steel.

### 3.5. Intra-Granular Acicular Ferrite Nucleation Induced by Non-Metallic Inclusion in HAZ

The published research shows that some special inclusions can serve as the nucleation core to form interlaced acicular ferrite (AF) with large difference in orientation, which can refine grains and improve HAZ toughness [27,28]. The formation of AF in HAZ can be ascribed to (i) effectively avoid the formation of large plate ferrite (mainly GBF) and strip ferrite which are perpendicular to the grain boundary (mainly WF) and the coarse residual M-A island, and (ii) the appropriate amount of inclusion particles meeting the condition of ferrite heterogeneous nucleation [4,26]. As shown in Figure 5 and Figure 6, in benchmark steel, after Al deoxidation, the non-metallic inclusions in the HAZ are mainly composite inclusions with Al_2_O_3_ as the core and MnS covering the surface. After Mg treatment, the oxide has evolved into Al–Mg–O + MnS with increased numbers and reduced size. In general, the inclusion-induced AF nucleation is mainly determined by thermal cycling conditions (cooling rate in the temperature range of 500–800 °C), the prior austenite grain size, and the inclusion characteristics [26]. Thus, it is necessary to explore the influence of different typical inclusions on the ferrite nucleus during the submerged arc welding process.

Figure 12 shows the typical mutual position of Al_2_O_3_ and ferrite in CGHAZ in welded benchmark steel. It can be seen that there are two types of ferrites around this Al_2_O_3_ inclusion: one is coarse polygonal ferrite (PF), the other is fine PF whose length–width ratio is less than 0.6 (α_1_ and α_2_ in Figure 12a). This indicates that, after the submerged arc welding procedure, this Al_2_O_3_ inclusion is prone to locate inside coarse PF rather than the refined AF. Meanwhile, it was also found that some AF exists next to the MnS of the Al_2_O_3_ + MnS composite inclusions, but the sizes of these AFs are still small, as shown in Figure 12b. Reference [29] indicated that AF and PF had different fracture characteristics. When cracks propagate through AF, the stress concentrated at the front of the crack is reduced by the deformation, and the crack expands in a wavy shape, forming a tearing dimpled fracture with high impact toughness. On the contrary, because it is hard to attain compatible deformation between PF and the surrounding microstructures, the cracks are more likely generated at the phase boundary and expand through ferrite as cleavage fractures by forming cleavage steps on the fracture surface. The morphology of PF corresponds to the fracture appearance and its impact toughness is generally lower than that of AF [25]. Thus, it can be speculated that for benchmark steel, after submerged arc welding thermal cycle, the austenite grains should be coarsened seriously. Due to the weak function of Al_2_O_3_ inclusions on promoting AF nucleation, during the cooling procedure, ferrite is preferred to initially precipitate at the austenitic grain boundary, and eventually grow into large plate ferrite (mainly GBF) and strip ferrite perpendicular to the grain boundary (mainly WF) rather than the fine AFs, deteriorating the toughness at low temperature.

As analyzed above, no coarse GBF or WF were found in the CGHAZ of Mg-treated steel after W1 or W2 procedures. However, a certain amount of AF could be identified. It is thus speculated that the high impact toughness of HAZ in the Mg-treated steel is strongly related to the AF. Figure 13 shows the typical position relationship between Al–Mg–O (noted as A and B) and ferrite in CGHAZ in Mg-treated steel after the W2 procedure. The mapping results show that Al and Mg are uniformly distributed in the inclusion. MnS is on the surface of the Al–Mg–O inclusion. The size of the inclusion is about 1.5–2.0 μm. It can be seen from Figure 13 that there are four strip-like ferrites (α_3_, α_4_, α_5_, α_6_) around inclusion A. Both α_3_ and α_5_ ferrites nucleate on the surface of this inclusion. α_4_ and α_6_ are sympathetic nucleation ferrites which are formed on the surface of α_3_ and α_5_, respectively. There are mainly five strip-like ferrites (α_1_, α_2_, α_6_, α_7_, α_8_) around inclusion B. Three ferrites (α_1_, α_2_, α_6_) are nucleations and growths depending on the inclusion B directly. α_7_ and α_8_ are sympathetic nucleation ferrites which are formed on the surface of α_1_ and α_6_, respectively. The co-inducing effect leads to the formation of interlaced and interlocked AF, effectively avoiding the formation of coarse GBF and WF. It is obvious that the cross-nucleated ferrite will significantly refine the HAZ structure and improve the welding properties of the steel.

It needs to be pointed out that PF can also be found around some Al–Mg–O+MnS. As shown in Figure 14, PF is identified in the MnS zone of the composite inclusion Al–Mg–O+MnS; meanwhile, acicular ferrite (α_1_) is observed in Al–Mg–O. This indicates that even under the similar thermal cycling conditions, the same inclusions may also have different effects on ferrite nucleation. Despite this, it was found that most of the Al–Mg–O+MnS inclusions introduced by Mg addition can indeed induce the AF formation in CGHAZ during the submerged arc welding heat cycle.

The microstructures of FGHAZ and ICHAZ in No.1 and No.2 steel (Figure 7 and Figure 8) are PF with a small amount of pearlite, which indicates that under this thermal cycle, both Al_2_O_3_ and Al–Mg–O are not likely to induce AF nucleation (Figure 15). This might mainly be because in addition to inclusion, the thermal cycle condition (e.g., mainly the cooling rate in the temperature range of 500–800 °C) is also an important factor affecting the inclusion-induced AF nucleation based on the literature [26,29]. Similarity, our previous research also confirmed that for Mg-treated steel, AF could only be observed when the cooling rate was in the range of 1.0–20 °C/s. When the cooling rate is around 5 °C/s, the AF is dominant. Thus, it is believed to have negative effect on promoting the nucleation of inclusion-induced AF when the cooling rate is too high or too low. In the real welding thermal cycle procedure, the peak temperatures of both FGHAZ and ICHAZ are all (<1200 °C) significantly lower than that of fusion zone and CGHAZ [1]. It is thus speculated that the reason for the insignificant AF nucleation in these two regions is the cooling rate, which may be not suitable for AF nucleation. Nevertheless, it can be identified clearly that the grain size of ferrite in these two regions in Mg-treated steel is smaller than that in the steel without Mg treatment (Table 6). This might mainly be because Mg influences refining of the grain size of the prior austenite grain. In order to verify this speculation, a confocal laser scanning imaging (CLSM) method was adopted to perform the in situ observation on the growth of austenite grains in the two experimental steels at 1200 °C. The results are shown in Figure 16, and for detailed experimental methods related to CLSM, please refer to our previously published papers [30]. It is identified from the figure that when the temperature rises to 1200 °C, the austenite grain size of the steel with Mg and without Mg treatment are 114.24 ± 18.43 μm and 55.36 ± 5.86 μm, respectively. It is believed that Mg would refine the prior austenitic grain size and improve the grain uniformity. Therefore, the refined microstructure in FGHAZ and ICHAZ in Mg-treated steel should be ascribed to the smaller prior austenitic grain at peak temperature.

### 3.6. Mechanisms of AF Nucleation Induced by Non-Metallic Inclusion in HAZ

The above analysis indicated that most of the inclusion nuclei for AF are the composite phases by oxide + MnS, with a size between 1.5 and 3.0 μm. The characterization on typical inclusion-induced AF shows that AF is more likely to nucleate on (i) the place nearby the MnS on the oxide surface, or (ii) the surface of an oxide such as Al–Mg–O.

The Mn-depleted zone (MDZ) mechanism is one of the most widely accepted explanations on MnS-induced AF nucleation. Usually, the MDZ mechanism is adopted to explain the positive effect of Ti_2_O_3_ on ferrite nucleus [31]. According to the MDZ mechanism, Mn is austenite stabilization element; the sharp drop of Mn concentration around inclusion will reduce the austenite phase stability and make the ferrite nucleation become a priority [32]. Byun et al. [33] pointed out that when the concentration of Mn decreased from 1.6 to 0.6%, A_e3_ increased from 830 to 862 °C, and the ferrite nucleation driving force at 700 °C increased from 320 to 380 J/mol.

According to the literature review, there are mainly two explanations for the formation of MDZ around Ti_2_O_3_ particles: (i) the precipitation of MnS on Ti_2_O_3_ under a cooling process leads to MDZ forming around it [34], (ii) there are many cation vacancies in Ti_2_O_3_ which will absorb Mn atoms. The untimely diffused Mn atoms around Ti_2_O_3_ lead to Mn-deficient zone [27,35]. In this research, Mn was not identified in the oxide core of the composite inclusions although it was found on the surface of the inclusions, which makes the precipitation of MnS explanation a more likely occurrence during the steel solidification. Wakoh et al. [36] found that when the content of sulfur in the steel was lower than 100 ppm, MnS would form on the surface of some specific oxides. When the S content increased to be higher than 100 ppm, almost all the oxides in the steels can be seamed as a MnS formation nucleus. Lee and Tomita et al. [37,38] found that the optimum S content for significant AF nucleation under MDZ mechanism is around 50 ppm. It can clearly be identified that the S content in experimental steel is generally 30~60 ppm, which is close to the optimal range. Even though the composite inclusion in this study was not Ti_2_O_3_, based on our previous research results, a narrow low-Mn zone can be found around MnS precipitated on the surface of the oxide [7]. This can be used to reasonably explain the formation of AF induced by the MnS of Al_2_O_3_ + MnS and Al–MgvO+MnS composite inclusions.

The mechanism based on the theory of minimum mismatch can be used to explain the ferrite nucleation in oxide regions in this research. It is believed that a heterogeneous nucleation can only be promoted when the crystal structure and lattice constant of the new phase are all perfectly matched to the parent phase [39]. Table 8 shows the crystal structure of the related inclusions obtained from the Inorganic Crystal Structure Database (ICSD). It can be found that Al–Mg–O and MnS belong to a cubic system, and Al_2_O_3_ belongs to a trigonal system. This suggests that Al–Mg-O and α-Fe are in the same crystal system, while Al_2_O_3_ and α-Fe are quite different. Based on these, it can be speculated that during the solidification procedure, the weaker inducing behavior of Al_2_O_3_ on AF formation is greatly related to the larger lattice mismatch between it and the matrix.

The lattice mismatch between the new phase and the parent phase can be distinguished by the lattice disregistry, based on Equation (2) which is proposed by Bramfitt [40]. The disregistry can be calculated based on the atom distances in the new phase and parent phase if they have the same crystal lattice.
(2)$$\delta =\frac{\left|d_{1}-d_{2}\right|}{d_{1}}\times 100\%$$
where *d*_1_ is the average distance of the atoms in the parent phase, and *d*_2_ is the atom distance of the new phase. When the disregistry is less than 6.0%, the parent phase is very effective for the new phase nucleation; when it is between 6.0% and 12%, the parent phase has a moderate effect on new phase nucleation; whereas when it is greater than 12%, the parent is least effective for the new phase nucleation.

The disregistry between the Al–Mg–O, MnS and α–Fe was calculated. Here, taking MgAl_2_O_4_ as an example, the bond distances of O–O, O–Mg and O–Al in MgAl_2_O_4_ are 2.8829 Å, 1.9319 Å and 1.9402 Å, respectively. When ferrite begins to nucleate, due to the strong affinity between Fe and O, Fe atoms are more likely to accumulate around O atoms in this situation. Furthermore, the atom distance of O–O (2.8829 Å) in MgAl_2_O_4_ and Fe–Fe (2.8665 Å) in α–Fe are quite similar, leading to a much smaller misfit, promoting the phase transformation. Based on the theory of Bramfitt [40], it can be speculated that α–Fe prefers to nucleate on the crystal surface with the O–O bond, and grows into an AF strip along certain directions. The calculated results of the disregistry between the related inclusions and α–Fe are compiled in Table 9. It can be seen that the lattice disregistry of different Al-Mg–O inclusions and α–Fe are all smaller than 2.0%, significantly lower than the standard (<6.0%) proposed by Bramfitt [40]. It is thus confirmed that the nucleation behavior of Al-Mg–O–induced AF in this situation can be well explained by the best disregistry theory. Similarly, the disregistry between MnS and α–Fe can also be calculated by comparing the Mn–Mn atom distance (4.8950 Å) and Mn–S atom distance (2.4475 Å) in MnS with the Fe–Fe atom distance in α–Fe. The disregistry is about 17%, clearly larger than 6.0%, which evidences that the induced AF nucleated around by MnS is not determined by the disregistry mechanism but by the MDZ mechanism.

## 4. Conclusions

After submerged arc welding, the microstructure in the fusion zone and CGHAZ of the 0.0026%Mg-treated ship plate steel is mainly AF and GF, while in FGHAZ and ICHAZ, the dominated microstructure is PF mixed with a small amount of pearlite. The grains in Mg-treated steel are much finer compared with those in the untreated steel. It shows that the impact toughness of the HAZ in Mg-treated steel is better than that of the benchmark steel under 29.55–44.11 kJ/cm heat input, and the impact toughness stability of the Mg-treated sample is significantly higher. The micro-hardness test results indicate that with Mg treatment, the hardness of the fusion zone, CGHAZ, and FGHAZ are all decreased to some extent. It is believed that Mg treatment might reduce the cold cracking tendency of the weak welding zone;After 26 ppm Mg treatment, the inclusions in HAZ changed from Al_2_O_3_ + MnS into Al–Mg–O+MnS, the percentage of the inclusions whose size are smaller than 2.5 μm increased from 90% (benchmark steel) to 98%, and the corresponding total number of the inclusions increased from 1268 to 2204 in the view field of 5.936 mm^2^. Under the welding heat inputs of 29.55 kJ/cm and 44.11 kJ/cm, in the steel without Mg treatment, Al_2_O_3_ + MnS hardly induced AF nucleation in CGHAZ, while in the Mg-treated steel, it was much easier for Al–Mg–O+MnS to induce AF nucleation in the same region. In FGHAZ and ICHAZ, both Al_2_O_3_+MnS and Al–Mg–O+MnS hardly promoted AF formation. Mg has an effect on refining the prior austenite grain size leading to a much finer microstructure in FGHAZ and ICHAZ in Mg-treated steel compared with benchmark steel;In CGHAZ, it was found that the induced AF can nucleate on the surface of the oxide, and also at the places near the MnS which is formed on the oxide surface. The formation behavior of Al–Mg–O+MnS-induced AF nucleation can be well explained by the Mn-deficiency zone mechanism and the lowest misfit mechanism.

## Figures and Tables

**Figure 1 materials-14-02445-f001:**
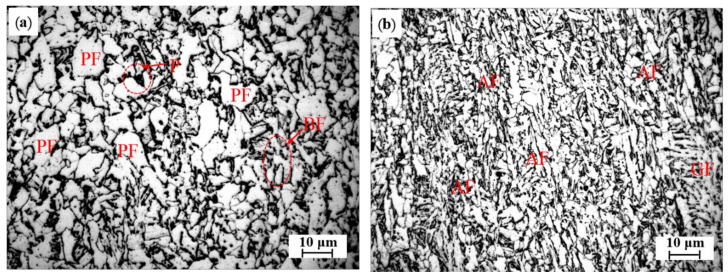
Optical micrographs of the rolled microstructure of steels: (**a**) No.1; (**b**) No.2, and the PF is polygonal ferrite; AF is acicular ferrite; BF is bainitic ferrite; GF is granular bainite ferrite.

**Figure 2 materials-14-02445-f002:**
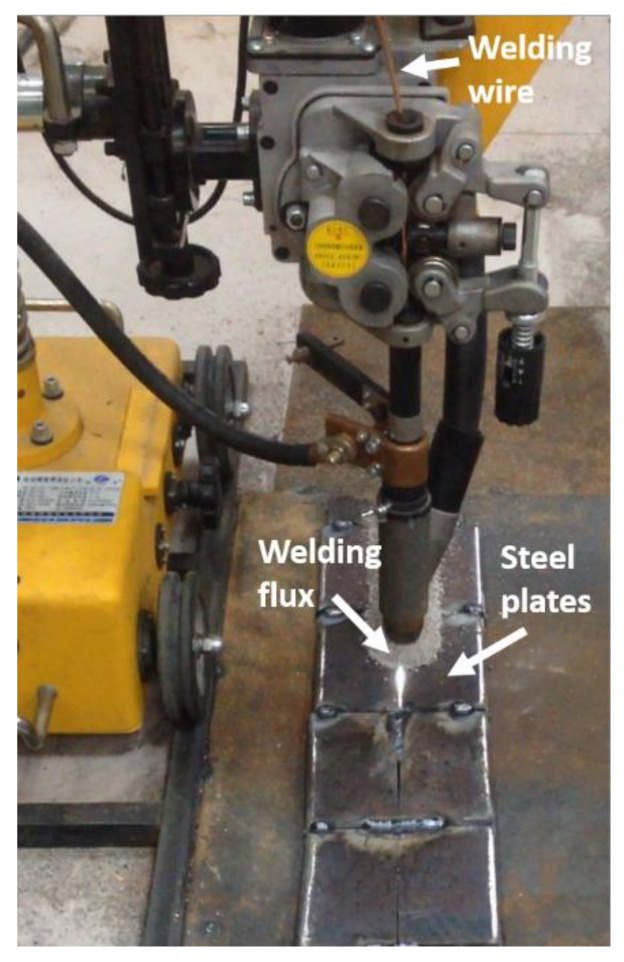
Picture of the welding equipment.

**Figure 3 materials-14-02445-f003:**
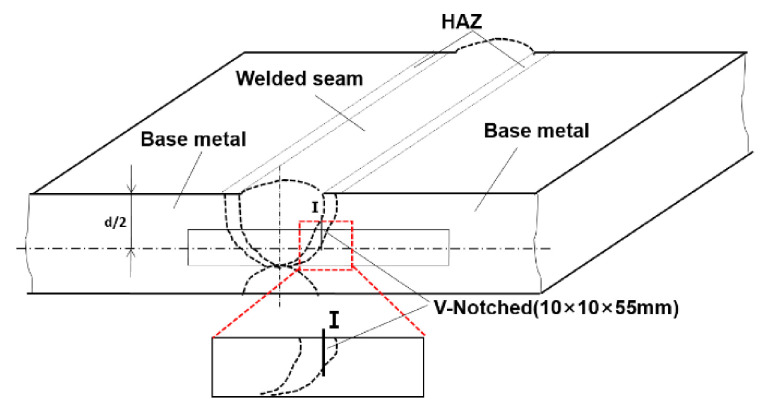
Position of micro-hardness measurement.

**Figure 4 materials-14-02445-f004:**
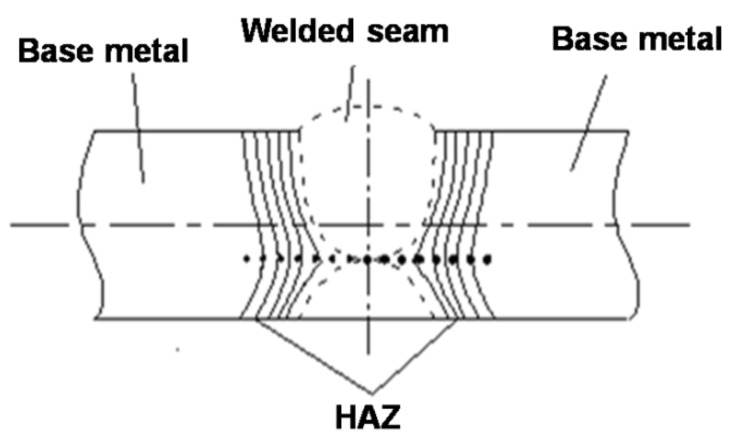
Position of micro-hardness measurement.

**Figure 5 materials-14-02445-f005:**
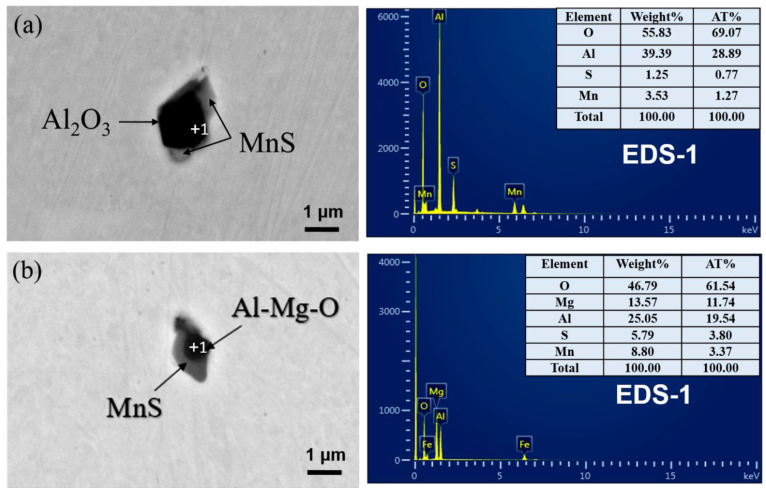
Morphology and composition of inclusions in HAZ of (**a**) benchmark steel and (**b**) Mg-treated steel.

**Figure 6 materials-14-02445-f006:**
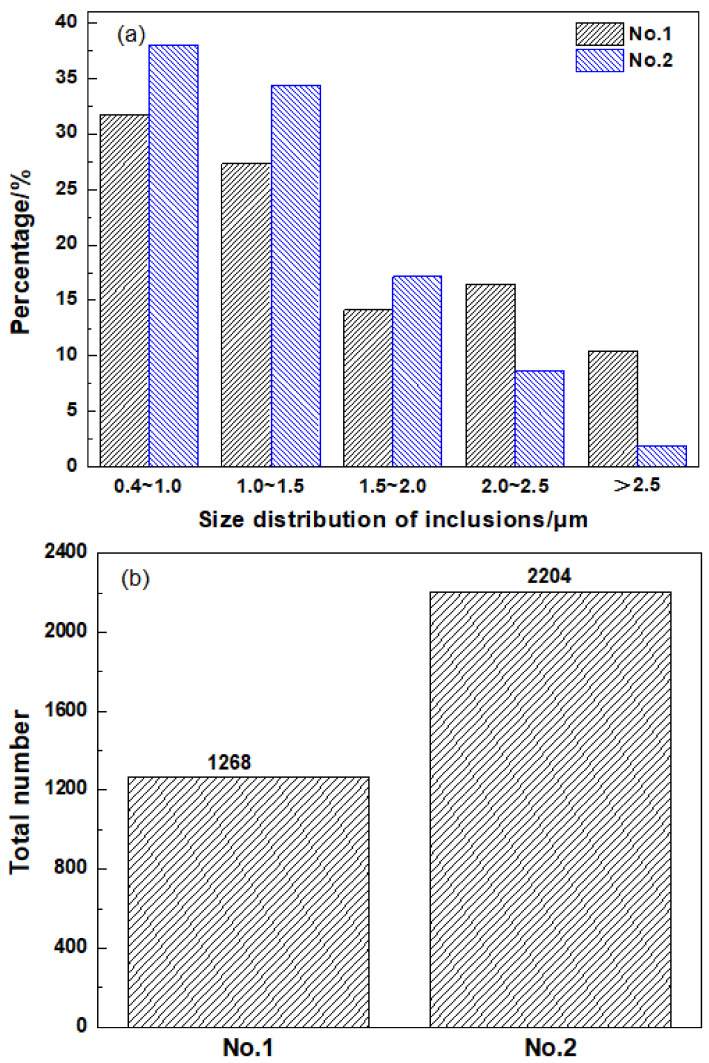
Size and number of inclusions in HAZ. (**a**) Non-metallic inclusion particle size distribution and (**b**) total number of nonmetallic inclusions.

**Figure 7 materials-14-02445-f007:**
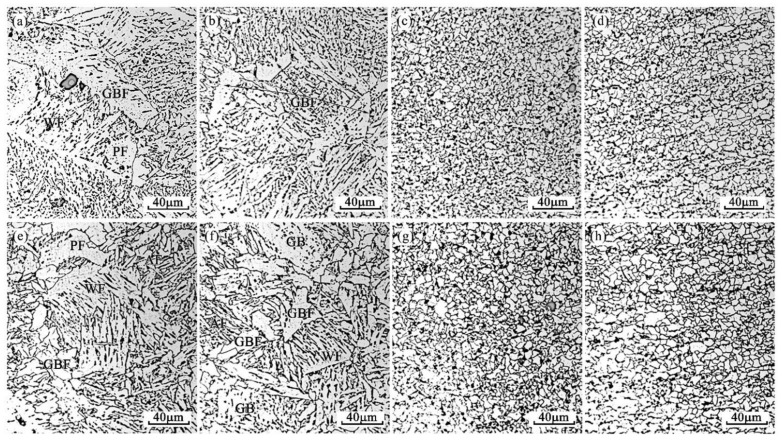
Microstructures of HAZ for No.1 steel. (**a**–**d**) CGHAZ, FGHAZ and ICHAZ under W1 procedure, (**e**–**h**) CGHAZ, FGHAZ and ICHAZ under W2 procedure.

**Figure 8 materials-14-02445-f008:**
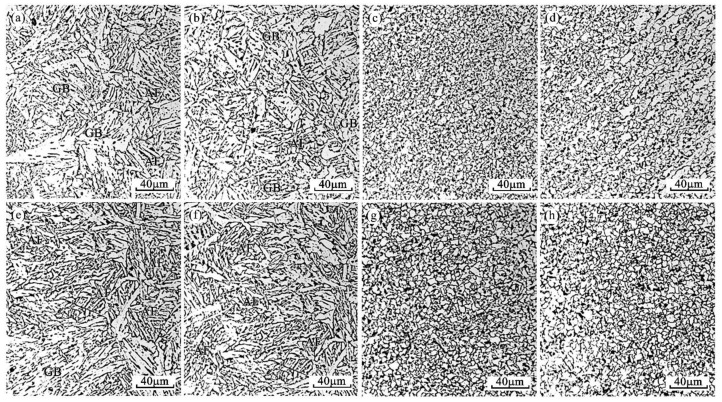
Microstructures of HAZ for No.2 steel. (**a**–**d**) CGHAZ, FGHAZ and ICHAZ under W1 procedure, (**e**–**h**) CGHAZ, FGHAZ and ICHAZ under W2 procedure.

**Figure 9 materials-14-02445-f009:**
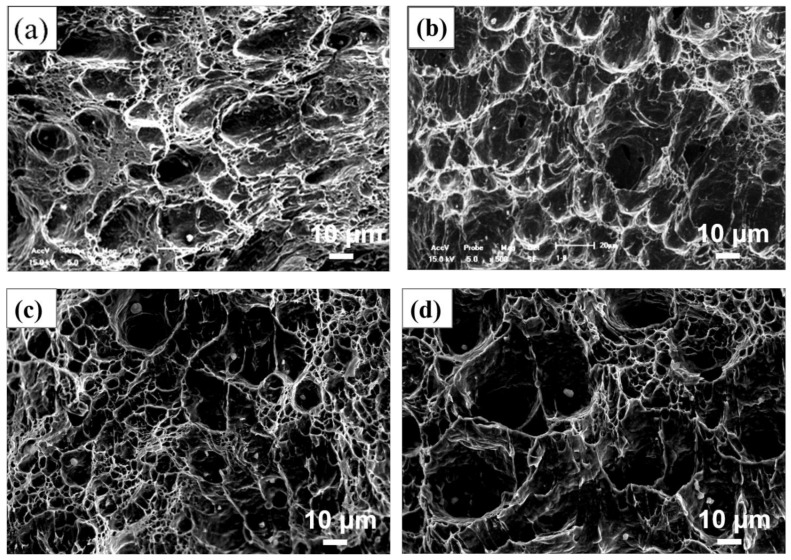
SEM fracture surfaces of steels under different welding processes. (**a**,**b**) No.1 steel under W1 and W2 welding process, (**c**,**d**) No.2 steel under W1 and W2 welding process.

**Figure 10 materials-14-02445-f010:**
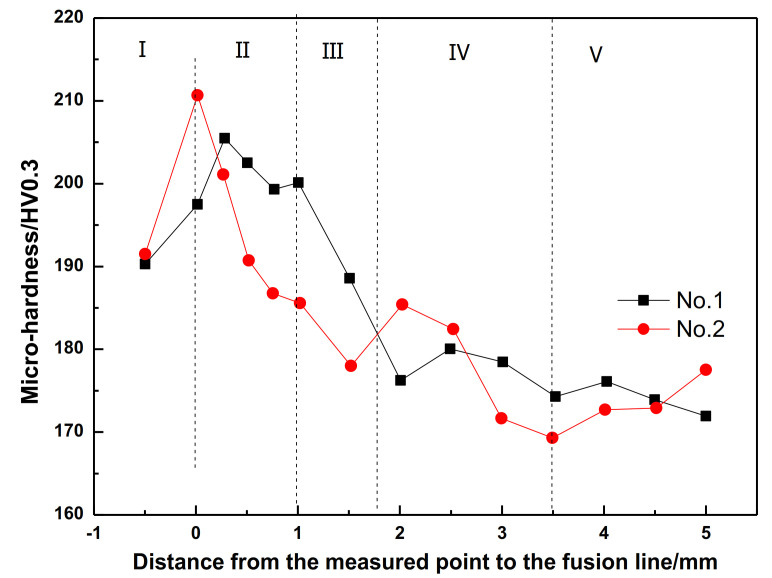
Micro−hardness variation from welding line to base metal for the two steels. The five regions (I–V) in the figure are weld line (I), fusion and CGHAZ (II), FGHAZ (III), ICHAZ (IV), and base metal (V).

**Figure 11 materials-14-02445-f011:**
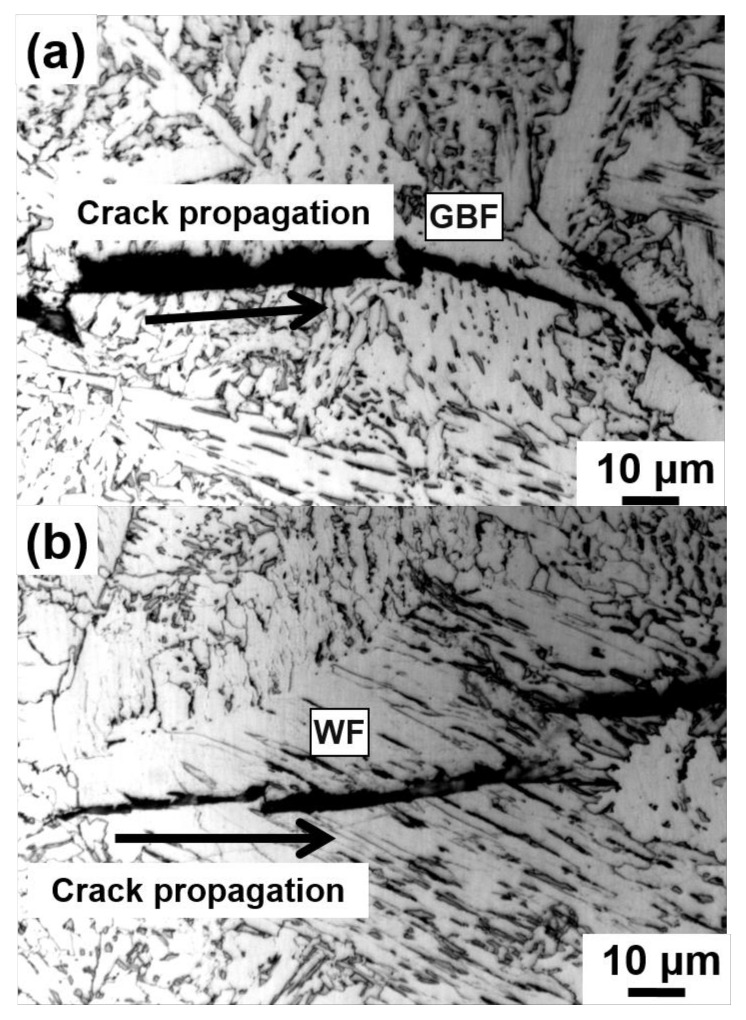
Propagation path for cleavage crack in the CGHAZ of the Mg-free steel (**a**) with grain boundary ferrite and (**b**) with widmanstätten ferrite.

**Figure 12 materials-14-02445-f012:**
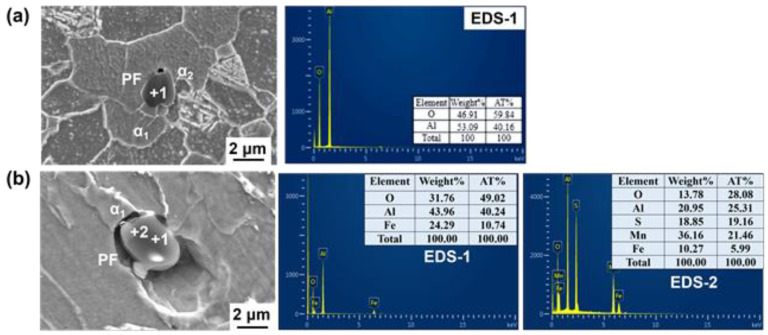
The relationship between inclusions and ferrite in the HAZ of No.1 steel: (**a**) Al_2_O_3_; (**b**) Al_2_O_3_+MnS.

**Figure 13 materials-14-02445-f013:**
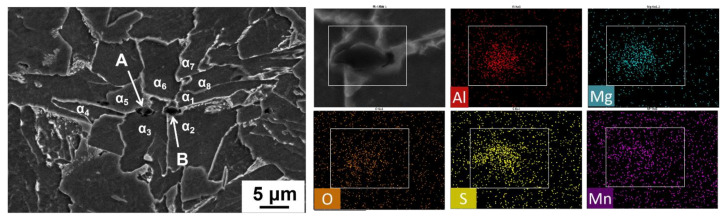
Mapping scanning results and the relationship between the Al–Mg–O+MnS and ferrite in the HAZ of Mg-treated steel at W2 heat input.

**Figure 14 materials-14-02445-f014:**
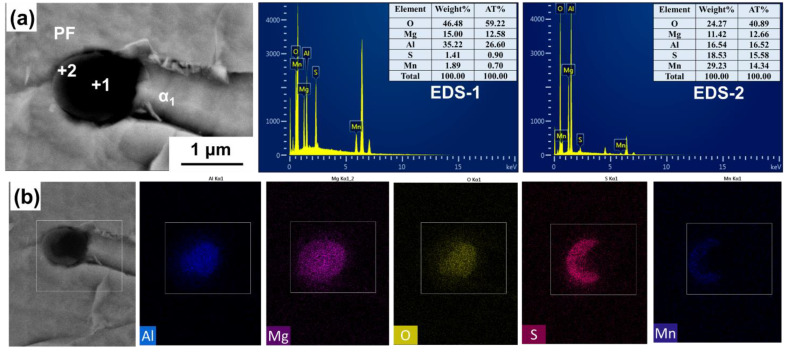
Mapping scanning results and the relationship between the Al–Mg–O+MnS and ferrite in the HAZ of Mg-treated steel at W2 heat input: (**a**) EDS analysis; (**b**) mapping analysis.

**Figure 15 materials-14-02445-f015:**
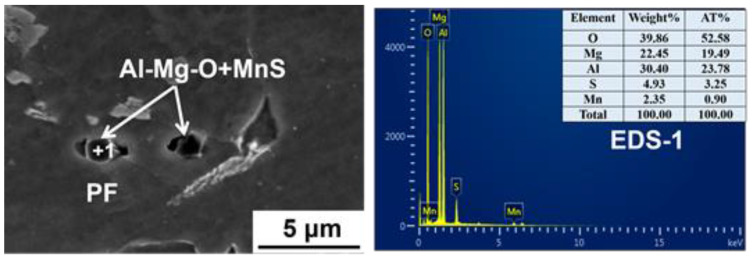
The relationship between Al–Mg–O+MnS inclusion and ferrite in the FGHAZ of No.2 steel.

**Figure 16 materials-14-02445-f016:**
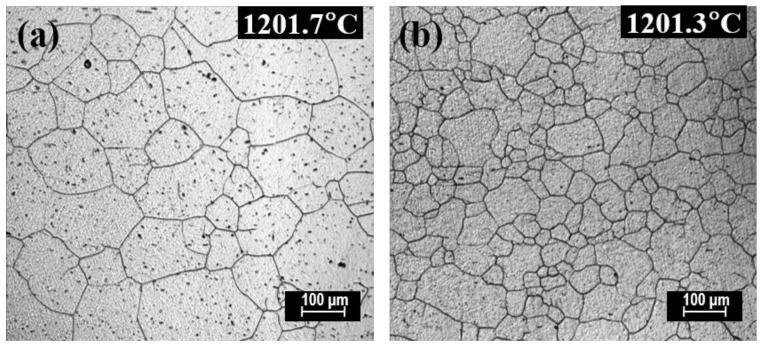
Austenite grains morphology after heating up to 1200 °C in the CLSM experiment: (**a**) No.1; (**b**) No.2.

**Table 1 materials-14-02445-t001:** Typical investigations concerning the effect of Mg addition on HAZ in past two decades.

Year	Authors	Metal	T.Mg/1 × 10^−6^	Typical Inclusions	Purpose *	Ref.
2004	Kojima et al.	low carbon steel	-	MgO, MgS, Mg (O,S)	B1,B2	[4]
2009	Chai et al.	low carbon steel	20–60	Ti–Mg–O	A1	[9]
2009	Shin et al.	API X70 linepipe steels	10	Ti–Mg–Al–Ca–O	A1	[10]
2011	Zhu et al.	low carbon steel	50	MgO	A1	[11]
2011	Zhu et al.	low carbon steel	50	MgO	A2	[12]
2011	Zhu et al.	low carbon steel	50	MgO	A2	[13]
2011	Zhan et al.	low carbon steel	100	Mg–Al–Zr–Ti–O	A3	[14]
2015	Yang et al.	EH36 shipbuilding steel	27–99	Ti–Mg–Al–O	A3	[15]
2017	Xu et al.	EH40 steel	15	Ti–Mg–O	A1	[16]
2018	Song et al.	0.15%C-1.31%Mn Steel	20	Ti–Mg–O	A3	[17]
2018	Li et al.	EH36 shipbuilding steel	50	Ti–Mg–Al–O	B1	[18]
2018	Lou et al.	EH420 steel	30	Ti–Ca–Mg–O	A1	[19]
2019	Li et al.	low carbon steel	26	Al–Mg–O	A1,A3	[20]
2019	Zou et al.	EH36 shipbuilding steel	7	Zr–Ca–Mg–O	A1	[21]
2019	Liu et al.	low carbon steel	50	Ti–Al–Mg–O	A2	[22]
2020	Xu et al.	EH36 shipbuilding steel	2–44	Ti–Mg–O	A3	[23]

* Note: A—effects on HAZ during the simulated weld process: acicular ferrite (A1), austenite grain growth (A2), inclusions (A3). B—effects on HAZ during the actual weld process: acicular ferrite (B1), austenite grain growth (B2).

**Table 2 materials-14-02445-t002:** Chemical composition of test steel, wt.%.

No.	C	Si	Mn	P	S	Ni	Al	Nb	Ti	N	O	Mg	Fe
1	0.052	0.23	1.53	0.009	0.003	0.29	0.028	0.040	0.014	0.0076	0.0037	-	Bal.
2	0.051	0.20	1.55	0.008	0.005	0.31	0.030	0.038	0.013	0.0065	0.0040	0.0026	Bal.

**Table 3 materials-14-02445-t003:** Chemical composition of the H10Mn2, wt.%.

Welding Wire	C	Si	Mn	P	S	Cr	Ni	Cu
H10Mn2	0.05	0.05	1.50	0.012	0.01	0.05	0.02	0.05

**Table 4 materials-14-02445-t004:** Flux chemical composition of SJ101 (wt.%).

Welding Flux	SiO_2_ + TiO_2_	CaO + MgO	Al_2_O_3_ + MnO	CaF_2_
SJ101	20~30	25~35	20~30	15~25

**Table 5 materials-14-02445-t005:** Process parameter of submerged arc welding.

Welding No.	Side	Welding Current/A	Welding Voltage/V	Welding Speed/m·h^−1^	Welding Speed/cm·s^−1^	Line Energy /kJ·cm^−1^
W1	front	545	31.1	30.4	0.844	20.08
	back	752	38.2	35.0	0.972	29.55
W2	front	545	31.7	30.5	0.839	20.21
	back	781	39.2	25.0	0.694	44.11

**Table 6 materials-14-02445-t006:** Microstructure characteristics of the different zones for the steel.

Welding No.	Line Energy/kJ·cm^−1^	Steel (Processing)	Structure types or PF Grain Size	Microstructure Characteristics			
				Fusion Zone	CGHAZ	FGHAZ	ICHAZ
W1	Front: 20.08 Back: 29.55	No.1 (Benchmark steel)	Structure types	GBF,WF,GF,BF	GBF,BF,GF	PF,P	PF,P
			grain size of PF/μm	-	-	4.88 ± 0.35	5.93 ± 0.83
		No.2 (Mg-treated steel)	Structure types	GBF,WF,GF,BF	GBF,BF,GF	PF,P	PF,P
			grain size of PF/μm	-	-	5.98 ± 1.15	6.40 ± 1.56
W2	Front: 20.21 Back: 44.11	No.1 (Benchmark steel)	Structure types	AF,GF	AF,GF	PF,P	PF,P
			grain size of PF/μm	-	-	3.67 ± 0.32	4.14 ± 0.61
		No.2 (Mg-treated steel)	Structure types	AF,GF	IAF,GF	PF,P	PF,P
			grain size of PF/μm	-	-	4.16 ± 0.35	4.22 ± 0.63

**Table 7 materials-14-02445-t007:** HAZ impact toughness for the steels after subjecting to submerged arc welding.

Welding No.	Samples	Test Values			Average Value ± SD
W1	No.1	292	285	259	279 ± 17
	No.2	248	237	245	243 ± 6
W2	No.1	300	305	307	304 ± 4
	No.2	301	289	268	286 ± 17

**Table 8 materials-14-02445-t008:** Crystal structure of the related inclusions (ICSD).

Inclusions	Crystal Structure	Lattice Constant(Å)		
		A	B	C
α-Fe	cubic	2.8665	2.8665	2.8665
Mg_0.4_Al_2.4_O_4_ (Al/Mg ≈ 6)	cubic	7.9736	7.9736	7.9736
Mg_2.175_Al_0.735_O_4_ (Al/Mg ≈ 3)	cubic	8.0405	8.0405	8.0405
MgAl_2_O_4_ (Al/Mg = 2)	cubic	8.1350	8.1350	8.1350
Al_2_O_3_	trigonal	4.7570	4.7570	12.988
MnS	cubic	4.8950	4.8950	4.8950

**Table 9 materials-14-02445-t009:** Results of the disregistry between the related inclusions and α-Fe.

Inclusions	O–O (Å)	O–Mg (Å)	O–Al (Å)	Misfit * (%)
Mg_0.4_Al_2.4_O_4_ (Al/Mg ≈ 6)	2.8212	1.8216	1.9399	1.6
Mg_2.175_Al_0.735_O_4_ (Al/Mg ≈ 3)	2.8476	1.8843	1.9309	0.66
MgAl_2_O_4_ (Al/Mg = 2)	2.8829	1.9319	1.9402	0.57

* Fe–Fe distance in α-Fe is 2.8665Å.

## Data Availability

The data presented in this study are available on request from the corresponding author. The data are not publicly available due to the large amount and variety of data that were processed.

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
