# Peer review of "Effect of Mg Addition on the Microstructure and Properties of a Heat-Affected Zone in Submerged Arc Welding of an Al-Killed Low Carbon Steel"

_materials, 2021, doi:10.3390/ma14092445_

Round 1

Reviewer 1 Report

Effect of Mg Addition on Microstructure and Properties of  Heat-affected Zone in Submerged Arc Welding of a Al-killed Low Carbon Steel

Ref: materials-1176995

Corresponding author: X.B. Li, [email protected]

General: This paper presents an investigation of   the Mg addition on the AF nucleation in Al-killed low carbon steel (application to ship plate steel) during welding process. The comparison between reference test and actual Mg treated plate was systematically studied according the microstructure and property evolution of the fusion zone. It is clearly stated that Mg treatment can significantly reduce the cold cracking tendency of the welding weak zone. Finally the authors propose an interesting discussion on the mechanisms (MDZ and minimum disregistry theories) underlying the AF nucleation.

The paper is well written with a concise and clear presentation. The authors follow the requirement of the Materials editor by dividing the paper into 3 main sections (materials and methods, results and discussion). A very clear conclusion gathers the main results of this experimental study. The figures are suitably introduced along the paper, they are well drawn and clear. The authors make a pedagogic effort to bring all the comments for a excellent understanding of the research work. In conclusion, this is a very good and very well written article on this topic.

Comments and minor revisions:

  • An appropriate figure or picture of the welding equipment would be appropriate in the section 2.2.
  • The thermal efficiency coefficient is assumed to be equal to 1 (Line 114). Please, give an explanation or at least a reference.
  • Chemical composition are always given in wt.% For some values, and particularly for trace elements, the ppm units would be more readable. This can be applied all along the paper.
  • The authors do not give a crystallographic analysis of the spinel inclusion Al-Mg-O. Actually, spinel inclusions (I think that even the word “spinel” is not introduced) must have a structure such as (MgO)x Al2O3. The reader must wait for the last section and Table 5 to get the appropriate structure of spinel. This could be an important improvement.
  • We can regret that the effect of thermal history (thermal cycle) experienced by the HAZ in welding operation is not deeply investigate. A schematic temperature diagram of the thermal cycle with the precipitation mechanisms reported on it would benefit the analysis.
  • The section 3.5 analyses the inclusion population in the HAZ for the two samples (without or with Mg trace). Is there any specific population of inclusion in the HAZ compared to the base metal? In other words, does the thermal cycle in welding operation change/modify the inclusion population (nature, number density)?

Very few language correctness or improvements of the text:

Line 105  “The heat input”  : instead use: “The welding line energy q/v “

Line 142  “measure by” : “measured by”

Line 154: “the content of oxygen is about 0.004 wt.%” : “the total oxygen content is about 40 ppm” . because the oxygen analysis include the dissolved oxygen (a short fraction but none zero) and the precipitates) .

Line 175: “in only 15 second [6].” : “in only 15 seconds [6]”

Line 408: “nucleation become a priority” : “nucleation to become a priority”

Line 483-484: “and the total number of the inclusions larger than 0.4 mm increased from” : “and the corresponding total number of the inclusions increased from”. Because Mg treatment reduces the nb of inclusions larger than 2.5 mm.

Author Response

Responses to reviewers’ comments

Dear editors and reviewers:

Thank you very much for your professional comments and suggestions on our previous manuscript entitled “Effect of Mg Addition on Microstructure and Properties of Heat-affected Zone in Submerged Arc Welding of a Al-killed Low Carbon Steel” (Research paper, No. materials-1176995). Those comments are really helpful to improve the quality of our article and also deepen our understanding of the scientific laws behind experimental phenomena.

We have carefully studied all the comments in sequence and made corresponding corrections according to the reviewers’ suggestions and/or requirement. And all the modifications are highlighted in the revised manuscript. We believe that the results are relatively new and should be of interest to the Mg effect in steel and also hope the revised revision could meet your expectation and get your approval. Thank you again for giving us the opportunity to revise and improve our manuscript. Below are our point-by-point responses to the reviewers’ comments.

Sincerely

Pro. Xiaobing Li

Reviewer 1

General: This paper presents an investigation of   the Mg addition on the AF nucleation in Al-killed low carbon steel (application to ship plate steel) during welding process. The comparison between reference test and actual Mg treated plate was systematically studied according the microstructure and property evolution of the fusion zone. It is clearly stated that Mg treatment can significantly reduce the cold cracking tendency of the welding weak zone. Finally the authors propose an interesting discussion on the mechanisms (MDZ and minimum disregistry theories) underlying the AF nucleation.

The paper is well written with a concise and clear presentation. The authors follow the requirement of the Materials editor by dividing the paper into 3 main sections (materials and methods, results and discussion). A very clear conclusion gathers the main results of this experimental study. The figures are suitably introduced along the paper, they are well drawn and clear. The authors make a pedagogic effort to bring all the comments for a excellent understanding of the research work. In conclusion, this is a very good and very well written article on this topic.

The question is(Question 1): An appropriate figure or picture of the welding equipment would be appropriate in the section 2.2.

Response 1: Thank you for your careful work and professional advice. In the revised manuscript, the picture of the welding equipment used in this study has shown in Figure 2

Question 2The thermal efficiency coefficient is assumed to be equal to 1 (Line 114). Please, give an explanation or at least a reference.

Response 2: Thank you for your kind advice. In this revised manuscript, we have given a reference that is also considered the thermal efficiency coefficient to be equal to 1.

Question 3Chemical composition are always given in wt.%. For some values, and particularly for trace elements, the ppm units would be more readable. This can be applied all along the paper.

Response 3: Thanks very much for your careful review. We have checked the manuscript, and all the chemical compositions are given in wt.%, and some trace elements, for example S, Mg, have given in ppm.

Question 4The author do not give a crystallographic analysis of the spinel inclusion Al-Mg-O. Actually, spinel inclusions (I think that even the word “spinel” is not introduced) must have a structure such as (MgO)x Al2O3. The reader must wait for the last section and Table 5 to get the appropriate structure of spinel. This could be an important improvement.

Response 4: Thank you for your suggestion. In order to make the reader understand easily, we have given the structure of (MgO)x Al2O3 when the (MgO)x Al2O3 inclusion is first introduced in the Introduction.

Question 5We can regret that the effect of thermal history (thermal cycle) experienced by the HAZ in welding operation is not deeply investigate. A schematic temperature diagram of the thermal cycle with the precipitation mechanisms reported on it would benefit the analysis.

Response 5: Thank you for your kind advice. In this study, we mainly focus on the function of Mg on the microstructure and properties of a ship plate steel after subjected to double-side submerged arc welding based on the acicular ferrite formation in different areas of HAZ. And the possible precipitation mechanisms during welding may be considered in our future study.

Question 6The section 3.5 analyses the inclusion population in the HAZ for the two samples (without or with Mg trace). Is there any specific population of inclusion in the HAZ compared to the base metal? In other words, does the thermal cycle in welding operation change/modify the inclusion population (nature, number density)?

Response 6: Thank you for your kind advice. The inclusion characteristics in the Mg-free and Mg-added base metals were investigated in our previous research(J. Iron. Steel Res. Int., 2016, 23, 415-421.). Therefore, in this revised paper, the inclusion characteristics difference between base metals and HAZ is discussed, as following:

In comparison with the inclusion characteristics of as-casted base metal reported in our previous research[8], it can be seen that the present thermal cycle in welding operation does not change the inclusion types for the two steels, while the percentage of inclusions with size below 2.5 μm appears to be increased and the total number of inclusions also tends to be more after subjected to the welding process.

Reviewer 2 Report

The article is very interesting, but I would like to highlight a few points:

  1. Line 80-84 and Table 2: What was the method of measuring the content of N and O in the tested steels? In the case of the above elements, the use of spectral methods usually involves a large measurement error. How to justify such a high content of N in both steels, especially when using the vacuum method for their smelting?
  2. Line 100-101: It is suggested to include the chemical composition and selected mechanical properties of the H10Mn2 + SJ101 weld metal in the manuscript. Sugeruje się zamieszczenie w manuskrypcie składu chemicznego i wybranych właściwości mechanicznych stopiwa H10Mn2+SJ101.
  3. Line 106: The use of the phrase "experimental steel plates were very thin" seems unfortunate. Thin sheets are usually up to 3 mm thick. However, as I understand it, 13 mm thick sheets were used to make the welded joints?
  4. Line 128: The text stated that three samples were used for the tests, for which standard deviations were later determined. These results cannot be found in the manuscript. In addition, due to the small number of samples, it is suggested to include all individual impact test results. This will allow the potential reader to interpret the test results independently, which is quite important in the case of special processes (welding).
  5. Line 205/221: The microphotographs presented in Figs. 6 and 7, in the manuscript received for review, are of poor quality. Please pay attention to this in your final paper.
  6. Line 236: The obtained values of energy absorbed at the temperature (-60°C) are very high. Is there no mistake in this case? This is an order of magnitude more than the properties of the weld metal (https://www.czhucheng.com/h10mn2eh14-submerged-arc-welding-wire-product/) and much more than the previously obtained results for the base metal at (-20°C) (line 92) .
  7. Line 246: In my opinion, the statement "... Mg-added steel could possess a better impact toughness ..." is too far-fetched. The presented results of fractographic analysis do not justify such a conclusion. Morphologically, the fractures of both steels are different, i.e. they occurred under different load conditions. On the basis of ASM Handbook - Fractography and S. Kocańda - Fatigue Failure of Metals, the fracture of first steel was formed by shearing (the dimples are elongated due to the variable speed of crack propagation), and the second one by tension (uniform dimples formed at a relatively low fracture speed and when the direction of the main stress is approximately normal to the fracture surface). This state of affairs is most likely due to the inability to precisely determine the place where the notch was made in the impact test sample or from different areas of the fracture surfaces that were tested. Therefore, on the basis of the presented research results, it is not possible to infer the effect of Mg addition on the morphology of the fracture. Moreover, the area in which the research was carried out was not given. It can be assumed that it was the area of ​​the bottom of the mechanical notch (directly under the notch), that is in, HAZ (CGHAZ)?
  8. Line 269: Fig. 10 shows the hardness distributions of samples made according to the W2 scheme. Due to the considerations carried out by the authors on the impact (apart from the addition of Mg) of welding energy, it is recommended to include in the manuscript the hardness distributions also for the W1 process.
  9. Line 294-297: In the caption of Fig. 11, the statement regarding the presence of cleavage cracks was used. From the documentation provided (Fig. 11 and the fractographic images shown in Fig. 9), they do not indicate the occurrence of such a phenomenon. The transcrystalline cracks shown in Fig. 11 are most probably secondary cracks formed in the last phase of the cracking process. Therefore, they are not the main cracks and their presence cannot be related to the general cracking mechanism of both tested steels. Such a position could be verified after conducting a fractographic analysis in the same areas of fractures as shown in Fig. 11.

Author Response

Responses to reviewers’ comments

Dear editors and reviewers:

Thank you very much for your professional comments and suggestions on our previous manuscript entitled “Effect of Mg Addition on Microstructure and Properties of Heat-affected Zone in Submerged Arc Welding of a Al-killed Low Carbon Steel” (Research paper, No. materials-1176995). Those comments are really helpful to improve the quality of our article and also deepen our understanding of the scientific laws behind experimental phenomena.

We have carefully studied all the comments in sequence and made corresponding corrections according to the reviewers’ suggestions and/or requirement. And all the modifications are highlighted in the revised manuscript. We believe that the results are relatively new and should be of interest to the Mg effect in steel and also hope the revised revision could meet your expectation and get your approval. Thank you again for giving us the opportunity to revise and improve our manuscript. Below are our point-by-point responses to the reviewers’ comments.

Sincerely

Pro. Xiaobing Li

Reviewer 2

Question 1Line 80-84 and Table 2: What was the method of measuring the content of N and O in the tested steels? In the case of the above elements, the use of spectral methods usually involves a large measurement error. How to justify such a high content of N in both steels, especially when using the vacuum method for their smelting?

Response 1: Thank you for your careful work. In this revised manuscript, we have given the method of measuring the content of N and O in the tested steels, and this method is usually used to analyze the content of N and O in steel. The modified section as following:

The contents of O and N was determined by TC-600 oxygen-nitrogen analyzer, and the spectral analysis results of the steel ingots were compiled in Table 2, where No. 1 is the benchmark steel, No. 2 is the Mg-treated steel.

Question 2Line 100-101: It is suggested to include the chemical composition and selected mechanical properties of the H10Mn2 + SJ101 weld metal in the manuscript.

Response 2: Thank you for your kind advice. In this revised paper, we have added the chemical compositions of the H10Mn2 and SJ101 in Section 2.2, as following:

Table 3. Chemical composition of the H10Mn2, wt.%

C

Si

Mn

P

S

Cr

Ni

Cu

H10Mn2

0.05

0.05

1.50

0.012

0.01

0.05

0.02

0.05

Table 4. Flux chemical composition of SJ101 (wt/%)

SiO2+TiO2

CaO+MgO

Al2O3+MnO

CaF2

SJ101

20~30

25~35

20~30

15~25

Question 3 Line 106: The use of the phrase "experimental steel plates were very thin" seems unfortunate. Thin sheets are usually up to 3 mm thick. However, as I understand it, 13 mm thick sheets were used to make the welded joints?

Response 3: Thank you for your careful suggestion. In fact, the "experimental steel plates were very thin" is drawn based on the thick comparison between experimental steel plates and engineered steel plates(which are usually as large as 50mm or above this value). In this revised paper, we have given some necessary explanation about this, as following:

Since the experimental steel plates were relatively thinner than the engineered steel plates, when the heat input applied to the back-side was more than 44 kJ/cm, the steel plate was burn-through.

Question 4Line 128: The text stated that three samples were used for the tests, for which standard deviations were later determined. These results cannot be found in the manuscript. In addition, due to the small number of samples, it is suggested to include all individual impact test results. This will allow the potential reader to interpret the test results independently, which is quite important in the case of special processes (welding).

Response 4: Thank you for your kind advice. In this revised paper, we have added the all individual impact test results in Table 7. As following:

Table 7. HAZ impact toughness for the steels after subjecting to submerged arc welding

Samples

Test values

Average value±SD

W1

No.1

292

285

259

279±17

No.2

248

237

245

243±6

W2

No.1

300

305

307

304±4

No.2

301

289

268

286±17

Question 5Line 205/221: The microphotographs presented in Figs. 6 and 7, in the manuscript received for review, are of poor quality. Please pay attention to this in your final paper.

Response 5: Thank you for your kind advice. In this revised paper, we have improved try our best the quality of these two images.

Question 6Line 236: The obtained values of energy absorbed at the temperature (-60°C) are very high. Is there no mistake in this case? This is an order of magnitude more than the properties of the weld metal (https://www.czhucheng.com/h10mn2eh14-submerged-arc-welding-wire-product/) and much more than the previously obtained results for the base metal at (-20°C) (line 92) .

Response 6: Thank you for your careful work. Indeed, the previous paper, we have given the transverse impact toughness, but the impact properties of the HAZ are measured as the longitudinal impact toughness. So, in this revised paper, we have given the longitudinal impact toughness of these two samples, and it can be seen that the impact properties both the base metal and HAZ are located as a similar order of magnitude. As following:

The mechanical properties of these two steel plates are as follows: yield strength 448 MPa (No. 1), 473 MPa (No. 2); tensile strength 545 MPa (No. 1), 605 MPa (No. 2); elongation 33.6 % (No. 1), 36.5 % (No. 2), which were also reported in our previous paper[7]. And average longitudinal impact toughness (-60 ) are measured as 246 J (No. 1), 261 J (No. 2).

Question 7Line 246: In my opinion, the statement "... Mg-added steel could possess a better impact toughness ..." is too far-fetched. The presented results of fractographic analysis do not justify such a conclusion. Morphologically, the fractures of both steels are different, i.e. they occurred under different load conditions. On the basis of ASM Handbook - Fractography and S. Kocańda - Fatigue Failure of Metals, the fracture of first steel was formed by shearing (the dimples are elongated due to the variable speed of crack propagation), and the second one by tension (uniform dimples formed at a relatively low fracture speed and when the direction of the main stress is approximately normal to the fracture surface). This state of affairs is most likely due to the inability to precisely determine the place where the notch was made in the impact test sample or from different areas of the fracture surfaces that were tested. Therefore, on the basis of the presented research results, it is not possible to infer the effect of Mg addition on the morphology of the fracture. Moreover, the area in which the research was carried out was not given. It can be assumed that it was the area of the bottom of the mechanical notch (directly under the notch), that is in, HAZ (CGHAZ)?

Response 7: Thank you for your kind advice. Yes, after careful consideration, in this revised paper, we have deleted the "... Mg-added steel could possess a better impact toughness ...", and also pointed out the individual area of the fractures shown in Figure 8. As following:

Figure 8 shows the SEM morphology of impact fracture of No. 1 and No. 2 steel in crack-initiated area under W1 and W2 welding processes respectively. It can be clearly observed that under the W1 and W2 welding processes, the impact fracture of No. 1 and No. 2 steel is typical dimple-microvoid accumulation ductile fracture. It can be also found that the depth of impact fracture dimples is shallower in No. 1 steel than that of No. 2 steel under both W1 and W2 welding conditions, which further shows that the Mg-added steel could possess a better impact toughness.

Question 8Line 269: Fig. 10 shows the hardness distributions of samples made according to the W2 scheme. Due to the considerations carried out by the authors on the impact (apart from the addition of Mg) of welding energy, it is recommended to include in the manuscript the hardness distributions also for the W1 process.

Response 8: Thanks very much for your careful review. Owing to the limited samples, we did not carry out the hardness analysis of samples made according to the W1 scheme. But based on our previous research experience, the hardness distributions behavior for the present W1 and W2 scheme should be the similarity. Certainly, we will improve and supplement them in our future work.

Question 9Line 294-297: In the caption of Fig. 11, the statement regarding the presence of cleavage cracks was used. From the documentation provided (Fig. 11 and the fractographic images shown in Fig. 9), they do not indicate the occurrence of such a phenomenon. The transcrystalline cracks shown in Fig. 11 are most probably secondary cracks formed in the last phase of the cracking process. Therefore, they are not the main cracks and their presence cannot be related to the general cracking mechanism of both tested steels. Such a position could be verified after conducting a fractographic analysis in the same areas of fractures as shown in Fig. 11.

Response 9: Thank you for your kind advice. In this revised paper, we have made some necessary modifications based on the review’s suggestion to let the description more rigorous. As following:

Figure 10 shows the typical morphology of the secondary cracks along GBF and WF in the heat-affected zone of the Mg-free steel after W2 process. It can be seen that the secondary crack directly propagates through the GBF (Fig. 10(a)) and WF (Fig. 10(b)) without any impeding effects.

Round 2

Reviewer 2 Report

Dear Sirs,
Thank you very much for including my earlier comments in the manuscript. For this reason, I do not raise any more objections and predispose the work for publication.
Sincerely,
Łukasz Konat